# Carbon nitride supported Fe$_2$ cluster catalysts with superior performance for alkene epoxidation

Shubo Tian[1], Qiang Fu [2,3], Wenxing Chen [1,4], Quanchen Feng[1], Zheng Chen[1], Jian Zhang [1], Weng-Chon Cheong[1], Rong Yu [5], Lin Gu[6], Juncai Dong [7], Jun Luo [8], Chen Chen[1], Qing Peng[1], Claudia Draxl[2], Dingsheng Wang [1] & Yadong Li[1]

Sub-nano metal clusters often exhibit unique and unexpected properties, which make them particularly attractive as catalysts. Herein, we report a "precursor-preselected" wet-chemistry strategy to synthesize highly dispersed Fe$_2$ clusters that are supported on mesoporous carbon nitride (mpg-C$_3$N$_4$). The obtained Fe$_2$/mpg-C$_3$N$_4$ sample exhibits superior catalytic performance for the epoxidation of *trans*-stilbene to *trans*-stilbene oxide, showing outstanding selectivity of 93% at high conversion of 91%. Molecular oxygen is the only oxidant and no aldehyde is used as co-reagent. Under the same condition, by contrast, iron porphyrin, single-atom Fe, and small Fe nanoparticles (ca. 3 nm) are nearly reactively inert. First-principles calculations reveal that the unique reactivity of the Fe$_2$ clusters originates from the formation of active oxygen species. The general applicability of the synthesis approach is further demonstrated by producing other diatomic clusters like Pd$_2$ and Ir$_2$, which lays the foundation for discovering diatomic cluster catalysts.

[1] Department of Chemistry, Tsinghua University, 100084 Beijing, China. [2] Institut für Physik and IRIS Adlershof, Humboldt-Universität zu Berlin, 12489 Berlin, Germany. [3] School of Chemistry and Chemical Engineering, Shandong University, 250100 Jinan, China. [4] Beijing Key Laboratory of Construction Tailorable Advanced Functional Materials and Green Applications, School of Materials Science and Engineering, Beijing Institute of Technology, 100081 Beijing, China. [5] Beijing National Center for Electron Microscopy, School of Materials Science and Engineering, Tsinghua University, 100084 Beijing, China. [6] Institute of Physics, Chinese Academy of Sciences, 100190 Beijing, China. [7] Beijing Synchrotron Radiation Facility, Institute of High Energy Physics, Chinese Academy of Sciences, 100049 Beijing, China. [8] Center for Electron Microscopy, Tianjin University of Technology, 300384 Tianjin, China. These authors contributed equally: Shubo Tian, Qiang Fu, Wenxing Chen. Correspondence and requests for materials should be addressed to D.W. (email: wangdingsheng@mail.tsinghua.edu.cn)

Metal cluster catalysts at the sub-nanoscale often possess unique and unexpected catalytic properties that normally do not exist in the corresponding nanoparticle counterparts[1–9]. Upon deposition on a substrate, the few atoms in the catalysts could provide a compelling platform for bridging heterogeneous and homogeneous catalysis[10–16]. Since the nature of sub-nano systems can be significantly altered by adding or removing just one atom, gaining a deep understanding on the structure–property correlations is of great importance for designing catalysts with extraordinary activity and selectivity[17–19]. While the structures and compositions of sub-nano clusters can be well characterized by X-ray absorption fine structure (XAFS) spectra in conjunction with high-resolution electron microscopes[16,20,21], synthesizing monodispersed metal catalysts with atomic precision, which is the prime prerequisite, remains a great challenge.

Epoxides constitute important intermediates in fine chemical industry and biotransformation. In current processes for alkene epoxidation in liquids, an extensive use of expensive oxidants or large doses of co-reagents is usually required[22–24], which inevitably leads to an increase in the costs. To overcome this drawback, several homogeneous catalysts like iron- and ruthenium-substituted polyoxometalates have been developed, allowing $O_2$ to be the oxidant without a need for any co-reductant[25,26]. In contrast, corresponding heterogeneous catalysts are rarely reported. Supported sub-nano metal clusters, via bridging both types of catalysts, are thus expected to play a role in the reactions.

Herein, we employ a "precursor-preselected" wet-chemistry strategy to prepare $Fe_2$ clusters dispersed on an mpg-$C_3N_4$ substrate. The preselected metal precursors ensure the formation of diatomic clusters, whereas mpg-$C_3N_4$ provides abundant anchoring sites to stabilize the metallic species. The pyrolysis process is carefully optimized to guarantee a complete removal of organic ligands from the precursors, and at the same time,

prevent agglomeration of the $Fe_2$ clusters. The prepared $Fe_2$/mpg-$C_3N_4$ sample exhibits excellent catalytic performance toward epoxidation of *trans*-stilbene, which is absent when using iron porphyrin, single-atom Fe, or small Fe nanoparticles as the catalyst. First-principles calculations reveal that the unique reactivity of the $Fe_2$ clusters is attributed to the formation of active oxygen species. The synthesis approach reported in this work can be applied to produce other transition-metal dimers and paves the way for a precise design of nanocatalysts at the atomic scale.

## Results

**Synthesis and characterization of $Fe_2$/mpg-$C_3N_4$ samples.** Mesoporous graphitic carbon nitride was prepared following the previous literature[27]. The transmission electron microscopy (TEM) image (Supplementary Figure 1) and the X-ray diffraction (XRD) pattern (Supplementary Figure 2) demonstrate a graphitic packing structure of mpg-$C_3N_4$ with disordered spherical pores[27]. The infrared (IR) spectrum (Supplementary Figure 3) evidences the formation of extended C–N–C networks, according to the typical C–N heterocycle stretching modes in the region of ca. 1100–1600 $cm^{-1}$ and the breathing mode of tri-s-triazine units at about 810 $cm^{-1}$[28,29]. Regarding the produced $Fe_2$/mpg-$C_3N_4$ samples, there is no IR absorption peak that corresponds to the organic molecules in the $Fe_2$ precursor, indicating a complete removal of the ligands. The content of Fe is estimated to be about 0.15 wt% according to the inductively coupled plasma optical emission spectrometry (ICP-OES) analysis. Upon deposition of the $Fe_2$ clusters, the XRD pattern does not show any additional diffraction peaks of Fe (Supplementary Figure 2), and there are no obvious Fe particles in the TEM image (Supplementary Figure 4). These results serve as the first indication that $Fe_2$ clusters do not agglomerate during the pyrolysis process. The homogeneous distribution of the diatomic clusters is further supported

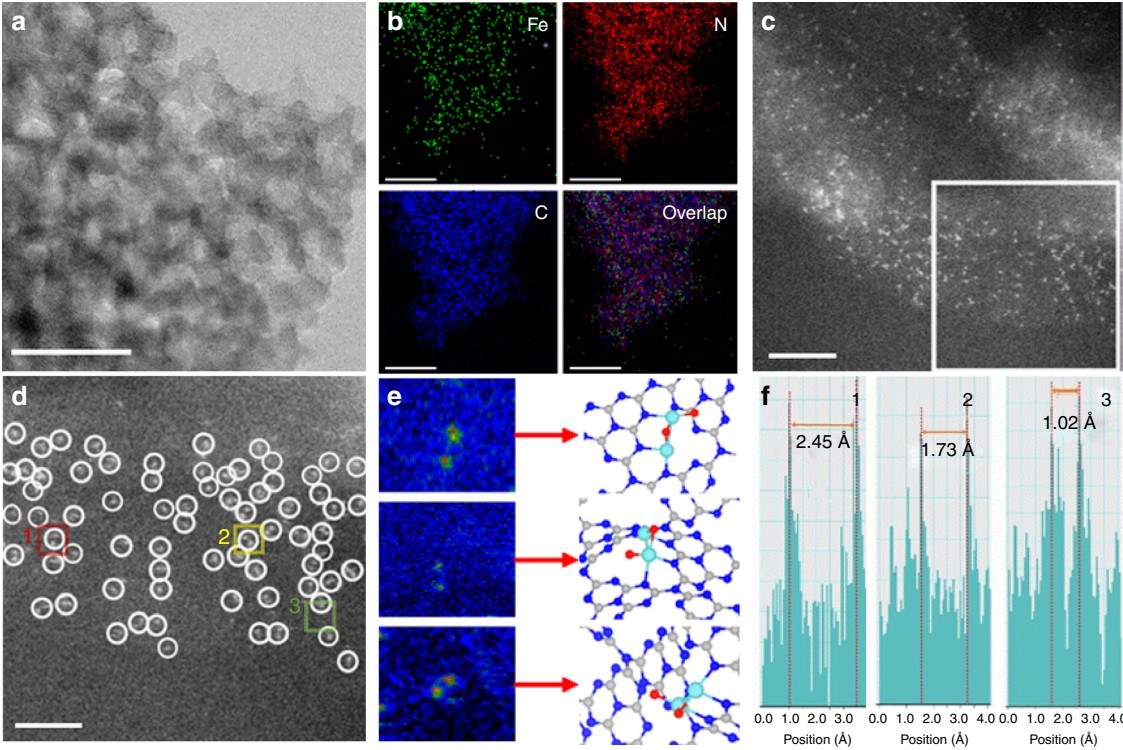

**Fig. 1** Characterization of $Fe_2$/mpg-$C_3N_4$ clusters. **a** HAADF-STEM images of $Fe_2$/mpg-$C_3N_4$. Scale bar, 50 nm. **b** Corresponding element maps showing distributions of Fe (green), N (red), and C (blue), respectively. Scale bar, 50 nm. **c** AC HAADF-STEM images of $Fe_2$/mpg-$C_3N_4$. Scale bar, 2 nm. **d** Magnified AC HAADF-STEM images of $Fe_2$/mpg-$C_3N_4$. Scale bar, 1 nm. **e**, **f** Intensity profiles obtained in areas 1, 2, and 3

by the HAADF-STEM image (Fig. 1a) and the corresponding energy dispersive X-ray (EDX) mapping analysis (Fig. 1b). The AC HAADF-STEM image (Fig. 1c) with atomic resolution further elucidates the characteristic of Fe atoms, where one can see the small bright dots homogeneously distributed on the mpg-$C_3N_4$ substrate. Due to the remarkable difference in Z-contrast between Fe and N/C[30], the small bright dots are determined to be Fe atoms. Furthermore, in the magnified AC HAADF-STEM image (Fig. 1d), a large proportion of isolated metallic diatoms appears in the regions tagged by white circles, confirming the formation of diatomic $Fe_2$ clusters. Since the AC HAADF-STEM image represents a two-dimensional projection along the incident beam direction, the detailed features of $Fe_2$ clusters are different from each other depending on their orientations in three dimensions[31]. For example, a group of bright double dots is consistent with a parallel $Fe_2$ structure, whereas a single bright dot corresponds to a $Fe_2$ dimer that is aligned with the projection. The statistical analysis on 100 pairs of $Fe_2$ dimers shows that the projected Fe–Fe distance between adjacent bright dots varies from 1.20 to 2.45 Å (Supplementary Figure 5). The largest distance, as shown in the intensity profiles (Fig. 1e, f), is consistent with the bond length of a $Fe_2$ dimer. When the support is replaced by graphene oxide, the $Fe_2$ clusters agglomerated to Fe nanoparticles (Supplementary Figures 6, 7) during the process of thermal decomposition. It is due to a lack of N atoms that can anchor the $Fe_2$ clusters. To further illustrate the $Fe_2$ site, we reduced the loading amount of $Fe_2$. The AC-STEM images show that the Fe atoms in the spherical electron microscope were still present as $Fe_2$ clusters, further indicating that the $Fe_2$ clusters did not decompose into single atoms during the synthetic process (Supplementary Figure 8). We also performed TOF-SIMS characterization of the samples. The data show that there is only $Fe_2$, but no larger Fe clusters, such as $Fe_3$ or $Fe_4$, indicating that the $Fe_2$ clusters did not agglomerate (Supplementary Figures 9, 10). The above results demonstrate the importance of the mpg-$C_3N_4$ support in the synthesis of the $Fe_2$ catalysts.

XAFS spectroscopy was utilized to probe detailed structure information such as the coordination environment[32]. Figure 2a shows the Fe k-edge X-ray absorption near-edge structure (XANES) spectra of the $Fe_2$/mpg-$C_3N_4$ sample compared with Fe Foil and $Fe_2O_3$ as references. The absorption edge of $Fe_2$/mpg-$C_3N_4$ is located between that of Fe Foil and $Fe_2O_3$, suggesting that the Fe atoms carry positive charges and are partially oxidized. The Fourier-transformed (FT) $k^3$-weighted EXAFS spectrum of the $Fe_2$ precursor is shown in Supplementary Figure 11. Here, a peak at a high $R$ value (ca. 2.50 Å) corresponds to the Fe–Fe coordination path. The other two peaks, at low $R$ values (ca. 1.82 and 2.14 Å), are assigned to the ligands of the $Fe_2$ precursor. Interestingly, these two peaks disappeared in the spectra of the $Fe_2$/mpg-$C_3N_4$ samples (Fig. 2b, c), meaning that the ligands had been completely removed. Figure 2b shows the FT $k^3$-weighted EXAFS spectrum of $Fe_2$/mpg-$C_3N_4$. At the FT curve of $Fe_2$/mpg-$C_3N_4$, a strong peak is located at ca. 1.53 Å, indicating that the sample is mainly comprised of the Fe–N/O coordination path. Interestingly, a secondary peak, which cannot be ignored at high $R$ value (ca. 2.27 Å), was found in the spectrum of the $Fe_2$/mpg-$C_3N_4$ sample. It reveals that some Fe–Fe path should also be accounted for as the surrounding coordination of metal centers. According to the EXAFS fitting results summarized in Fig. 2c, d, Supplementary Table 1, and Supplementary Figures 12–16, the average coordination numbers of Fe–N/O and Fe–Fe are 3.8 and 1.2, respectively. In order to further identify the local structure of $Fe_2$/mpg-$C_3N_4$, XANES and EXAFS simulations, which are very sensitive to the 3D arrangement of atoms around the photo-

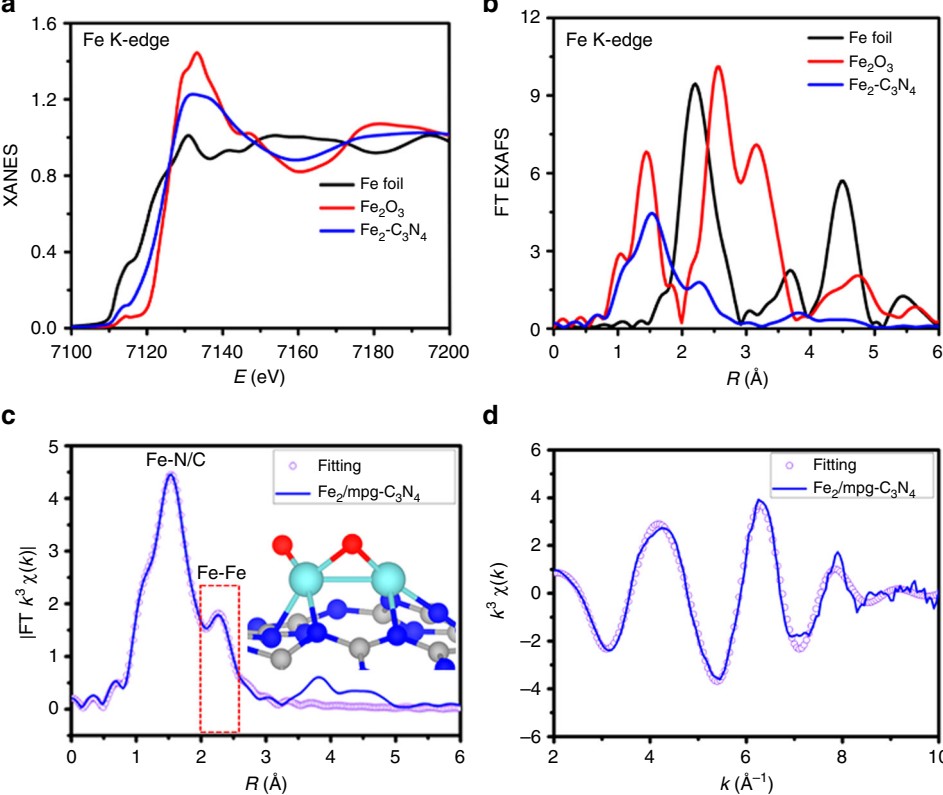

**Fig. 2** X-ray absorption analysis of Fe K-edge. **a** XANES spectra at the Fe k-edge of $Fe_2$/mpg-$C_3N_4$, $Fe_2O_3$, and Fe foil. **b** Fourier transform (FT) at the Fe k-edge of $Fe_2$/mpg-$C_3N_4$, $Fe_2O_3$, and Fe foil. **c**, **d** Corresponding fits of the EXAFS spectrum of $Fe_2$/mpg-$C_3N_4$ at R space and k space, respectively. The inset of **c** is the schematic model of $Fe_2$/mpg-$C_3N_4$ (Fe cyan, O red, N blue, and C gray)

absorber, were carried out at the Fe K-edge. Supplementary Figures 16 and 18 show that the simulated XANES and EXAFS spectrum based on our model agrees well with the experimental results, indicating that this structure is the most likely actual structure. A series of other possible structures were also considered, but the comparison between the simulated spectra and the experimental EXAFS and XANES results is quite unsatisfactory (Supplementary Figures 17 and 19), confirming this structure is the most likely actual structure. Combining the EXAFS fitting and the XANES simulations, the atomic structure of the Fe$_2$/mpg-C$_3$N$_4$ sample can be revealed. The structure of Fe$_2$/mpg-C$_3$N$_4$ was further identified by first-principles calculations (Fig. 2c, insets and Supplementary Figure 20). Here, Fe atoms are anchored by two N atoms in the graphitic carbon nitride framework. The two Fe atoms are slightly oxidized, connecting with two and one O atoms, respectively. The Fe–Fe bond length was calculated to be 2.40 Å, while it is around 2.2 Å without the presence of O atoms. Bader charge analysis revealed that the two Fe atoms possess positive charges of 1.32 and 1.00, respectively. The information based on DFT calculations agrees very well with results from the XAFS spectrum.

**Epoxidation of *trans*-stilbene to *trans*-stilbene oxide**. We next investigated the catalytic properties of the as-prepared Fe$_2$/mpg-C$_3$N$_4$ sample for epoxidation reactions. We chose *trans*-stilbene as the alkene reactant because of its non-volatility as well as the product stability for a reliable determination of conversion, yield, and selectivity using gas chromatography. Using molecular O$_2$ as the oxidant and without any additives, the Fe$_2$/mpg-C$_3$N$_4$ sample shows unique and superior catalytic performance toward the epoxidation.

As shown in Fig. 3a, we achieved conversion of 91% and selectivity of 93% after 24 h. It is one of the best results for the

epoxidation of *trans*-stilbene to *trans*-stilbene oxide using Fe-based catalysts, employing O$_2$ as the oxidant without any additive. When bare mpg-C$_3$N$_4$, iron porphyrin, or Fe nanoparticle (ca. 3 nm)/mpg-C$_3$N$_4$ were used (Supplementary Figures 21, 22), the *trans*-stilbene oxide product was almost undetectable under the same condition, uncovering the unique performance of the diatomic clusters. Single-atom catalysts have attracted much interest because of their remarkable catalytic activity, selectivity, and 100% atom utilization[33–36]. To further demonstrate the unique performance of Fe$_2$/mpg-C$_3$N$_4$, we synthesized the single-atom Fe$_1$/mpg-C$_3$N$_4$ sample for comparison. HAADF-STEM, AC HAADF-STEM, and XAFS (Supplementary Figures 23, 24) have confirmed that the as-prepared material contains only single atoms of Fe. When such sample was used, only trace amounts of *trans*-stilbene oxide product were obtained, confirming the unique and superior performance of the diatomic clusters. The performance of the Fe$_2$ clusters is also compared with that of other known catalysts. Noble metal nanoparticles (such as Ru, Rh, Pd, Au, and Pt) have been demonstrated to be good catalysts for epoxidation[37,38]. Herein, we synthesized a series of noble metal nanoparticles (Supplementary Figures 25–29). Supplementary Table 2 shows that the activity of Fe$_2$/mpg-C$_3$N$_4$ is much better than all the metal nanoparticles. We then collected the Fe$_2$/mpg-C$_3$N$_4$ catalyst after the reaction and reused it in the next round of epoxidation reaction. After 15 cycles, the Fe$_2$/mpg-C$_3$N$_4$ sample still maintains its pore structure and exhibits robust recycling capability with well-retained activity and selectivity (Fig. 3b). The unchanged structures as fresh samples identified by HAADF-STEM and AC HAADF-STEM images further corroborate the stability of the catalyst (Supplementary Figure 30).

First-principles calculations were performed to explore the underlying reasons for the unique catalytic properties of Fe$_2$/mpg-C$_3$N$_4$. In Fig. 3c, the energy profile for the epoxidation of

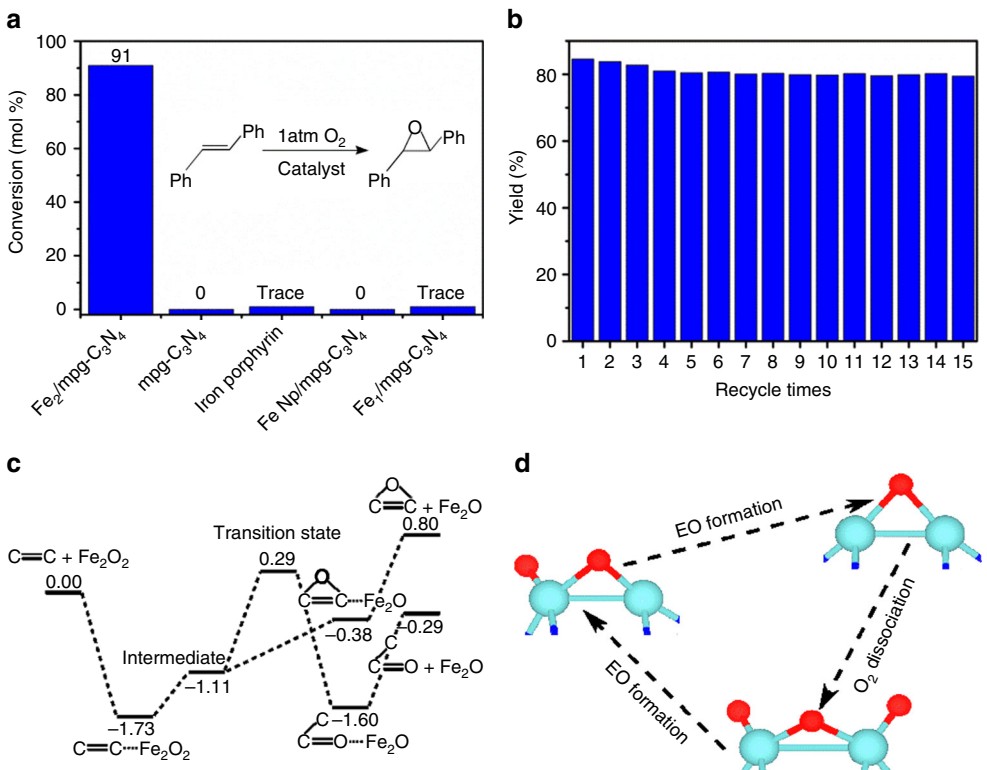

**Fig. 3** Epoxidation of *trans*-stilbene. **a** Catalytic epoxidation of *trans*-stilbene using different catalysts. **b** Recycle of Fe$_2$/mpg-C$_3$N$_4$ for catalytic epoxidation of *trans*-stilbene. **c** Energy profile (unit: eV) for the *trans*-stilbene epoxidation at the Fe$_2$O$_2$ site. **d** Consumption and regeneration of the active one-coordinated oxygen species

*trans*-stilbene at the $Fe_2O_2$ site is shown. We found that the alkene molecule first approaches the $Fe_2$/mpg-$C_3N_4$ catalyst via a non-planar configuration, where van der Waals interactions play an important role in the adsorption. Then, the molecule connects to the one-coordinated oxygen atom through one of the two carbon atoms in the C=C double bond, bringing about an intermediate state (Supplementary Figure 31) that controls the selectivity of alkene epoxidation[39–41]. After that, the other carbon atom is bonded to the one-coordinated oxygen atom, leading to the formation of the epoxidized product. Such process merely needs to overcome an energy increase of 0.73 eV. It is worth noting that the value is much lower than the energy barrier of 1.40 eV, which corresponds to a competitive pathway toward the formation of combustion products[39–41]. In the above process, only the one-coordinated oxygen atoms are active and play an important role, whereas the two-coordinated ones behave as bystanders. It is interesting to find that once one active oxygen species is consumed in the alkene epoxidation, two more one-coordinated oxygen atoms can be generated via $O_2$ dissociation at the $Fe_2O$ site (Supplementary Figure 32). The $O_2$ dissociation involves the formation of a highly activated molecular precursor and a significant energy release of 3.32 eV, which makes it a very facile process. In Fig. 3d, we schematically present the consumption and regeneration of the active oxygen species. Here, the catalytic activity and selectivity of the produced $Fe_2O_3$ are quite similar as those of $Fe_2O_2$, with the corresponding epoxidation profile shown in Supplementary Figure 33.

The striking difference in the catalytic activity of $Fe_2$/mpg-$C_3N_4$, Fe nanoparticles, and iron porphyrin is attributed to the variation of the interaction strength between $O_2$ molecules and the corresponding Fe species. For a good catalytic performance, such interaction should be neither too strong nor too weak[42,43].

On Fe nanoparticles, the interaction between their surface layers and the oxygen reactants is very strong[44,45], producing tightly bound oxygen atoms that can hardly participate in the epoxidation reaction. Regarding the iron porphyrin, by contrast, the interaction of $O_2$ with the embedded single Fe atom is too weak and only molecular adsorption is possible (Supplementary Figure 34). The bond length of 1.30 Å in the adsorbate, compared with that of 1.24 Å in isolated $O_2$, demonstrates that such molecule has not been well activated. Thus, neither Fe nanoparticles nor iron porphyrin can generate the active oxygen species as in the case of the $Fe_2$/mpg-$C_3N_4$ catalyst.

**Synthesis and characterization of other $TM_2$/mpg-$C_3N_4$ samples.** Some other transition-metal (for example, TM = Pd, Ir) clusters were produced using the same scheme, which demonstrates the general applicability of the approach for synthesizing diatomic clusters (Supplementary Figures 35–36). In the AC HAADF-STEM images (Fig. 4c, f, inset), a large proportion of bright double dots was observed in the regions tagged by white circles, indicating the existence of isolated metallic dimers in $TM_2$/mpg-$C_3N_4$. EXAFS spectrum was further used to confirm the as-obtained $TM_2$/mpg-$C_3N_4$ samples (Fig. 4a–f, Supplementary Figures 37–42, and Supplementary Table 3). In the FT $k^3$-weighted EXAFS spectrum shown in Fig. 4b, e, the first strong FT curves of TM k-edge in $TM_2$/mpg-$C_3N_4$ show peaks at 1.45 Å for $Pd_2$/mpg-$C_3N_4$ and 1.63 Å for $Ir_2$/mpg-$C_3N_4$ (before phase shift correction), indicating that $TM_2$/mpg-$C_3N_4$ is mainly comprised of the TM–N coordination path. Similarly, a secondary peak at high R value (2.35 Å for $Pd_2$/mpg-$C_3N_4$ and 2.49 Å for $Ir_2$/mpg-$C_3N_4$) was found, which demonstrates that metal–metal path should also be accounted for as in the case of $Fe_2$/mpg-$C_3N_4$.

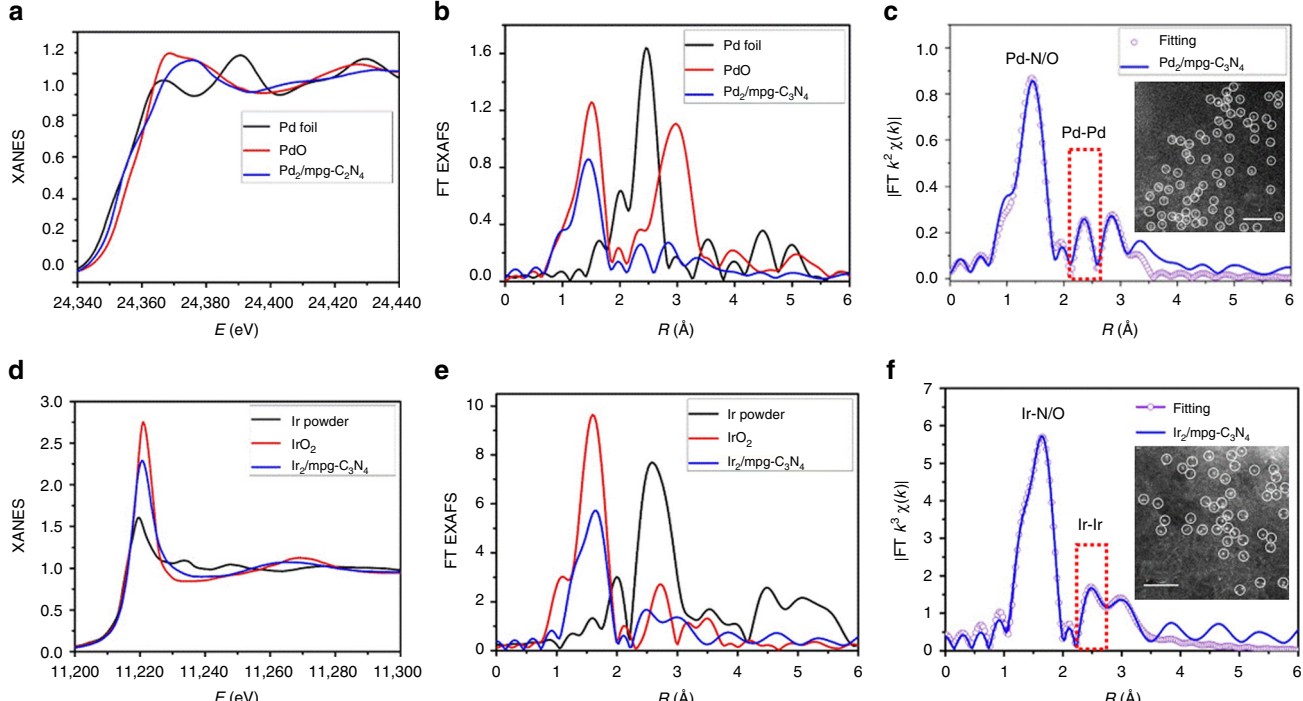

**Fig. 4** Characterization of other $TM_2$/mpg-$C_3N_4$ clusters. **a** XANES spectra at the Pd k-edge of $Pd_2$/mpg-$C_3N_4$, PdO, and Pd foil. **b** Fourier transform (FT) at the Pd k-edge of $Pd_2$/mpg-$C_3N_4$, PdO, and Pd foil. **c** Corresponding fits of the EXAFS spectrum of $Pd_2$/mpg-$C_3N_4$ at R space. The inset of (**c**) is the AC HADDF-STEM of $Pd_2$/mpg-$C_3N_4$, scale bar, 1 nm. **d** XANES spectra at the Ir $L_3$-edge of $Ir_2$/mpg-$C_3N_4$, $IrO_2$, and Ir powder. **e** Fourier transform at the Ir $L_3$-edge of $Ir_2$/mpg-$C_3N_4$, $IrO_2$, and Ir powder. **f** Corresponding fits of the EXAFS spectrum of $Ir_2$/mpg-$C_3N_4$ at R space. The inset of (**f**) is the AC HADDF-STEM of $Ir_2$/mpg-$C_3N_4$, scale bar, 1 nm

## Discussion

In summary, we have developed a "precursor-preselected" wet-chemistry strategy to synthesize $Fe_2$ clusters supported on mpg-$C_3N_4$, whose structures are identified using AC-STEM, XAFS, and first-principles calculations. By employing molecular oxygen as the oxidant and in the absence of aldehyde as co-reagent, the as-prepared $Fe_2$/mpg-$C_3N_4$ sample exhibits unique and superior catalytic performance toward alkene epoxidation. By contrast, iron porphyrin, single-atom Fe, and small Fe nanoparticles are nearly reactively inert. First-principles calculations reveal that the unique reactivity of $Fe_2$ comes from the formation of active oxygen species. Our scheme can be applied toward producing other diatomic clusters and establishes a substantial foundation for further studies of atomically precise sub-nano catalysts.

## Methods

**Preparation of mpg-$C_3N_4$.** A method was used according to a previous report with a tiny modification. Aliquot of 5 g of cyanamide and 12.5 g of colloidal silica Ludox HS-40 are mixed together until complete dissolution of cyanamide. The mixture was heated in an oil bath at 100 °C upon stirring for ca. 3 h until removal of water and formation of a white solid. The powder was then grounded in a mortar, transferred into a crucible, and heated under air at 2.3 °C min$^{-1}$ (4 h) up to 550 °C and then treated at 550 °C for 4 h. The as-obtained yellow powder was grounded in a mortar and then treated under stirring for 2 days in an $NH_4HF_2$ 4 mol L$^{-1}$ solution. The dispersion was then filtered, the precipitate washed with distilled water and ethanol. After filtering, the yellow compound is dried under vacuum at 100 °C overnight.

**Synthesis of $Fe_2$/mpg-$C_3N_4$.** In a typical synthesis of $Fe_2$/mpg-$C_3N_4$, 5 mg bis (dicarbonylcyclopentadienyliron) ($Fe_2O_4C_{14}H_{10}$) and mpg-$C_3N_4$ (500 mg) were dissolved in the DMF (100 mL) under stirring at room temperature for 24 h. The product was separated by centrifugation at 10,000 rpm for 5 min and washed subsequently with DMF for once, then washed with methanol for once and finally dried under vacuum at room temperature. The as-prepared powder was transferred into a ceramic broth and then placed into a tube furnace maintaining 300 °C for 2 h under flowing mixture of 5% $H_2$/Ar atmosphere with a heating rate of 5 °C min$^{-1}$. When the temperature is above 250 °C, thermogravimetric analysis (TGA) shows a weight loss of 30.8 wt%, similar to the theoretical loss of 31.5 wt% according to the formula when the ligand is removed completely (Supplementary Figure 43). Therefore, we chose 300 °C to thoroughly remove the organic ligands. The Fe loading is 0.15% determined by ICP-AES analysis.

*XAFS measurements and analysis.* The X-ray absorption fine structure spectra data (Fe k-edge and Ir $L_3$-edge) were collected at 1W1B station in Beijing Synchrotron Radiation Facility (BSRF, operated at 2.5 GeV with a maximum current of 250 mA). The X-ray absorption fine structure spectra data (Pd k-edge) were collected at 14W1 station in Shanghai Synchrotron Radiation Facility (SSRF, 3.5 GeV, 250 mA). The data were collected in fluorescence excitation mode using a Lytle detector. All samples were pelletized as disks of 13 mm diameter with 1 mm thickness using graphite powder as a binder. Using the ATHENA module in the IFEFFIT packages, we processed the acquired EXAFS data following the standard procedures. After the subtraction of the post-edge background and normalization, the EXAFS spectra was obtained. Then, the $\chi$(k) data were transformed to R space. Using the ARTEMIS module, we obtained the quantitative structural parameters via a least-squares curve parameter fitting method.

*Typical procedure for the epoxidation of* trans-stilbene. In this typical reaction, trans-stilbene (90.12 mg, 0.50 mmol), $Fe_2$/mpg-$C_3N_4$, iron porphyrin, or Fe nanoparticle/mpg-$C_3N_4$ (ca. 3 nm) (0.50 μmol Fe) and DMAC (5 mL) were mixed in a 20 mL of Schlenk tube. Then, we used an oil pump to remove the air in the tube. An $O_2$ balloon was used to blow about 1 atm $O_2$. Finally, the reaction vessel was heated in a silicon oil bath at 90 °C. The mixture was stirred at this temperature for 24 h. The products were identified by gas chromatography (GC).

**Computational details.** The mpg-$C_3N_4$ framework was simulated by a graphitic carbon nitride (g-$C_3N_4$) monolayer that exhibits a corrugated non-planar configuration. We adopted its optimized lattice constant of 6.937 Å and constructed a hexagonal 2 × 2 unit cell accordingly (Supplementary Figure 16). An $Fe_2$ dimer with different numbers of O atoms was deposited at various locations on the g-$C_3N_4$ substrate, in order to carry out an extensive structural exploration.

Spin-polarized density functional theory calculations were performed based on the projector-augmented-wave (PAW) approach[46], utilizing the Vienna ab initio simulation package (VASP)[47,48]. The energy cutoff of the plane-waves basis set to 500 eV. The exchange-correlation interactions were described by the optPBE-vdW functional[49,50]. The first Brillouin zone was sampled using a 3 × 3 × 1 Monkhorst–Pack grid[51]. Structural relaxations were performed until the maximum

residual force on each atom was <0.02 eV Å$^{-1}$. The transition state was located using the climbing image nudged elastic band method[52] with a force criterion of 0.05 eV Å$^{-1}$. A dipole correction to the total energies was applied along the vertical direction. Bader charge analysis was carried out with core charges included in the partitions[53]. All structures were visualized using the program VESTA[54].

**Data availability**. The data supporting this study are available from the authors on reasonable request.

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

## Acknowledgements

This work was supported by China Ministry of Science and Technology under Contract of 2016YFA (0202801), the National Natural Science Foundation of China (21521091, 21390393, U1463202, 21471089, and 21671117). Q.F. and C.D. acknowledge support from the German Research Foundation (DFG) and the European Union's Horizon 2020 research and innovation programme, Grant Agreement No. 676580 through the Center of Excellence NOMAD. Q.F. thanks supports from the Fundamental Research Funds of Shandong University (2018TB006). This work made use of the resources support of National Synchrotron Radiation Laboratory in Beijing and Shanghai.

## Author contributions

D.W. and Y.L. conceived and designed the research project. S.T. conducted and designed the experiments, analyzed the data, and wrote the paper. Q.F. and C.D. conducted the first-principles simulations and contributed to the writing of the paper. W.C. completed the XAFS characterization and corresponding data analysis. J.D. analyzed XANES data. Q.F., C.C., and Q.P. helped to analyze the data. L.G., R.Y., and J.L. provided spherical-aberration-corrected TEM techniques. Z.C. and J.Z. assisted to carry out the catalytic experiments. W.C. conducted HAAD-STEM. All authors contributed to the preparation of the manuscript.

## Additional information

**Competing interests:** The authors declare no competing interests.

