## [Peer Review File · Nature Communications]

Reviewers' comments:

Reviewer #1 (Remarks to the Author):

Using the “precursor-preselected” wet chemistry method, the authors successfully synthesized a unique carbon nitride Fe dimer catalyst for alkene epoxidation. A superior catalytic performance with 93% selectivity and 91% conversion for the epoxidation of trans-stilbene was observed. Although this work is interesting to the catalysis community, it is not very convincing and novel enough. Plus, there are still many technical points that need to be further clarified. The current version is not suitable to be published on Nature Commun.

1. It is not convincing to me that using two hydrogen atoms to replace the benzene ring in the DFT calculations is a good choice. I would like to believe there is a fundamental difference between the benzene ring is similar to the hydrogen because the C=C bond lengths is nearly the same (0.02 Å) for ethylene and trans-stilbene.
2. The dispersion interaction correction should be included in the DFT calculations, in particular, for the larger molecules with two benzene rings.
3. The side reaction pathways lead to the aldehyde formation should be included.
4. When it is dealing with a catalytic reaction, an entire cycle should be taken into account. In this work, two oxidation cycles for epoxidation of trans-stilbene should be explored.
5. It is not clear to me that how the O₂ dissociation over the Fe₂O? If the bridging oxygen o is not active, what the barrier for oxygen atom transform from the bridging to the terminal position?

Reviewer #2 (Remarks to the Author):

This work reports the catalytic conversion of stilbene to its epoxide by an Fe₂ catalyst. Conceptually, this idea is not new, because the organometallic community explored the metal complexes as precatalysts (followed by ligand removal) a long time ago (e.g., in the 1980s). But, technically, these authors have done a better job, esp. in AC STEM characterization. The majority of clusters seem to be Fe₂, though Fe₁ and larger ones cannot be fully avoided.

What about the single atom Fe catalyst? Without the control exp, it's hard to claim Fe₂ is unique and superior. Is the ligand removed for the iron porphyrin catalyst? If not, the catalytic activity would certainly be very different from that of single atom Fe₁ catalysts.

Compared to Fe NPs, normally the Fe₂ cluster would be oxidized more easily, because the Fe₂ is smaller and certainly more reactive. It is hard to understand that the Fe NPs get oxidized very easily and produce tightly bound O(ad) as the authors discussed. Why is that?

The Fe atoms are assumed to anchor to two N atoms in the graphitic carbon nitride support, but does X-ray absorption give any evidence for Fe-N bond formation? I'm not sure if Fe-N and Fe-O can be differentiated. If Fe₂ is anchored on O atoms, what would the results be?

I believe the m-C₃N₄ support is also important, though itself does not have high activity in the reaction. Can the authors do a comparison with graphene or GO with Fe₂ deposited as in m-C₃N₄?

Reviewer #3 (Remarks to the Author):

Small metal clusters mounted on solid supports are the subjects of growing attention and the focus of quite a lot of catalysis research.

The subject manuscript is another example, this one with iron clusters and a catalytic epoxidation reaction.

The work could be improved in a number of respects.

The identification of the iron clusters by STEM and XAFS should be stronger. The images show clusters of various sizes. The images should be subjected to statistical analysis of the sizes and the identification of the clusters as dimers should be examined critically.

The XAFS data analysis should include fits with various structural models and a statistically based selection of the best fitting model and a critical assessment of the comparison.

The assumption that the groups on the iron are removed in the treatment and not replaced by something else is glib. These are reactive species.

The catalytic data would have more meaning if they were compared with catalytic data for a good known epoxidation catalyst. The stability of the catalyst should be tested with flow reactor operation for a long enough period to allow a good assessment.

Responses to Reviewers' Comments:

Reply to the Report of Reviewer 1

Referee:

Using the “precursor-preselected” wet chemistry method, the authors successfully synthesized a unique carbon nitride Fe dimer catalyst for alkene epoxidation. A superior catalytic performance with 93% selectivity and 91% conversion for the epoxidation of trans-stilbene was observed. Although this work is interesting to the catalysis community, it is not very convincing and novel enough. Plus, there are still many technical points that need to be further clarified. The current version is not suitable to be published on Nature Commun.

Reply:

We appreciate the Referee’s comments that we address below point by point. In the revised version, first-principles calculations have been re-performed to resolve all the technical issues raised by the Referee. Besides, more experimental results have been added to further confirm the conclusions.

It may be worth noting that synthesizing supported sub-nano clusters with an exact atom number remains a challenge. In this work, a universal approach has been developed to prepare a series of transition-metal diatomic clusters, which can be directly identified using advanced characterization technique like aberration-corrected scanning transmission electron microscope. The unique and superior catalytic performance of the diatomic cluster has also been demonstrated by employing alkene epoxidation as an example. This work lays a foundation for designing heterogeneous catalysts with extraordinary reaction capabilities.

Referee:

1. It is not convincing to me that using two hydrogen atoms to replace the benzene

ring in the DFT calculations is a good choice. I would like to believe there is a fundamental difference between the benzene ring is similar to the hydrogen because the C=C bond lengths is nearly the same (0.02 Å) for ethylene and *trans*-stilbene.

Reply: We agree with the Referee that due to the different groups that are connected to the C=C double bond, there is a fundamental difference between ethylene and *trans*-stilbene, which is mainly reflected in two aspects: (1) The frontier orbitals of ethylene merely comes from the C=C double bond, whereas for those of *trans*-stilbene, contributions from the two benzene rings are non-negligible (Fig. R1). (2) The van der Waals interaction between the benzene rings and the mpg-C₃N₄ substrate is expected to play a role in the adsorption and reaction of *trans*-stilbene molecules. For instance, *trans*-stilbene adopts a non-flat configuration upon adsorption, which is not the same as in the case of ethylene.

However, it is worth emphasizing that the essence of the interaction between the C=C double bond and the Fe₂O_x species will be not changed upon the replacement of benzene rings in the computational model. Thus, the catalytic mechanism and the nature of the active sites revealed in the original manuscript are expected on firm ground.

Figure R1. The HOMO and LUMO orbitals of *trans*-stilbene and ethylene molecules.

In the revised version, we have re-performed all calculations using the *trans*-stilbene model and updated the manuscript accordingly. Despite the conclusions are still the same, the quality of the calculations has been significantly improved. We thank the Referee again for the insightful suggestions.

2. *The dispersion interaction correction should be included in the DFT calculations, in particular, for the larger molecules with two benzene rings.*

Reply: The Referee is absolutely correct that the dispersion interaction should be accounted when the adsorption and reaction of *trans*-stilbene is simulated by the DFT approach. In the revised version, all calculations are performed using the optPBE-vdW functional, which explicitly includes van der Waals interactions.

3. *The side reaction pathways lead to the aldehyde formation should be included.*

Reply: We have added calculations on the side reaction pathway toward 1,2-diphenylethanone (Note that it is no longer aldehyde since the hydrogen atoms have been changed to phenyl groups). Although such process is exothermic, it is kinetically hindered by an energy barrier as high as 1.40 eV, which explains the excellent selectivity of the catalysts.

In the revised manuscript, the above results and discussions have been included.

4. *When it is dealing with a catalytic reaction, an entire cycle should be taken into account. In this work, two oxidation cycles for epoxidation of trans-stilbene should be explored.*

Reply: We agree with the Referee that an entire cycle should be taken into account when dealing with a catalytic reaction. In the revised version, the epoxidation process of *trans*-stilbene on Fe₂O₂ and Fe₂O₃ species are both considered and included.

5. *It is not clear to me that how the O₂ dissociation over the Fe₂O? If the bridging oxygen o is not active, what the barrier for oxygen atom transform from the bridging to the terminal position?*

Reply: The process of O₂ dissociation over the Fe₂O site is shown in **Supplementary Fig. S28**. It can be seen that a molecular precursor (the middle panel) exists in the dissociative adsorption of O₂. In this state, the molecule is highly activated, with the O = O bond length increased from 1.24 Å to 1.43 Å. The energy release of 2.22 eV in this step could produce “hot” O atoms [*Phys. Rev. Lett.* **77**, 123 (1996)]. Moreover, the energy barrier of the subsequent O₂ dissociation is calculated to be quite small, which is less than 0.05 eV. Thus, the O₂ dissociation over Fe₂O, i.e., the regeneration of the active one-coordinated oxygen species, is a very facile process.

Figure S28. Dissociative adsorption of an O₂ molecule over the Fe₂O site. Numbers in black represent the bond lengths (unit: Å), whereas those in green refer to the energy changes. The formation of the molecular precursor (the middle panel) is barrierless, and the energy barrier of the subsequent O₂ dissociation is less than 0.05 eV.

Due to the different coordination number, the O atom at the bridge site (two-coordinated) is more stable than that at the terminal site (one-coordinated). Therefore, the first O atom that adsorbs at Fe₂ will be always at the bridge position, whereas only subsequent O atoms can locate at the terminal ones. In other words, the formation of a bridge O atom is the prerequisite for the formation of terminal O atoms. As a result, the bridge O atom cannot be transformed to the terminal position. If we place only one O atom at the terminal site, such atom will move to the bridge site after the atomic relaxation.

Reply to the Report of Reviewer 2

Referee:

This work reports the catalytic conversion of stilbene to its epoxide by an Fe₂ catalyst. Conceptually, this idea is not new, because the organometallic community explored the metal complexes as precatalysts (followed by ligand removal) a long time ago (e.g., in the 1980s). But, technically, these authors have done a better job, esp. in AC STEM characterization. The majority of clusters seem to be Fe₂, though Fe₁ and larger ones cannot be fully avoided.

Reply: We appreciate the Referee's comments that we address below point by point.

- 1. What about the single atom Fe catalyst? Without the control exp, it's hard to claim Fe₂ is unique and superior. Is the ligand removed for the iron porphyrin catalyst? If not, the catalytic activity would certainly be very different from that of single atom Fe₁ catalysts.*

Reply: We agree with the Referee that the catalytic performance of single-atom Fe₁ catalysts should be considered for comparison with that of the reported Fe₂ dimer in this work. Using iron porphyrin as the precursor and through the thermal decomposition method, we have successfully synthesized the single-atom Fe₁/mpg-C₃N₄ catalyst by removing the ligand of iron porphyrin. HAADF-STEM, AC HAADF-STEM and XAFS (**Supplementary Figure S19-20**) have confirmed that the as-prepared sample contains only single atoms of Fe. When the single-atom Fe₁/mpg-C₃N₄ sample was used, only trace amounts of *trans*-stilbene oxide product

was obtained, confirming the unique and superior performance of the diatomic clusters.

In a typical synthesis of $\text{Fe}_1/\text{mpg-C}_3\text{N}_4$, 20 mg iron porphyrin and 500 mg $\text{mpg-C}_3\text{N}_4$ were dissolved in 100 mL DMF under stirring at room temperature for 24h. The product was separated by centrifugation and then washed with DMF and methanol thoroughly. The as-prepared powder was pyrolyzed at 400 °C for 2h under flowing mixture of 5% H_2/Ar atmosphere, and during this period, the ligands of iron porphyrin were found to be removed. The Fe loading is 0.18% determined by the ICP-AES analysis.

Figure S19. (a) HAADF-STEM images of $\text{Fe}_1/\text{mpg-C}_3\text{N}_4$. (b) corresponding element maps showing distributions of Fe (green), N (red), C (blue), respectively. (c) AC HAADF-STEM images of $\text{Fe}_1/\text{mpg-C}_3\text{N}_4$. (d) Magnified AC HAADF-STEM images of $\text{Fe}_1/\text{mpg-C}_3\text{N}_4$.

Figure S20. (a) XANES spectra at the Fe k-edge of Fe₁/mpg-C₃N₄, Fe₂O₃, and Fe foil. (b) Fourier transform (FT) at the Fe k-edge of Fe₁/mpg-C₃N₄, Fe₂O₃ and Fe foil.

2. *Compared to Fe NPs, normally the Fe₂ cluster would be oxidized more easily, because the Fe₂ is smaller and certainly more reactive. It is hard to understand that the Fe NPs get oxidized very easily and produce tightly bound O(ad) as the authors discussed. Why is that?*

Reply: It seems that some sentences in the original manuscript had caused misunderstanding. We completely agree with the Referee that the oxidation of the Fe₂ cluster is more easily than that of the Fe NPs. In the original version, however, we did not attempt to compare the relative oxidation rate between Fe₂ cluster and Fe NPs. In fact, when we mentioned that the Fe NPs get oxidized very easily, it was intended to compare with the rate of stilbene epoxidation, meaning that the oxidation can indeed happen. In the revised version, we have rephrased the sentences for a clear expression.

3. *The Fe atoms are assumed to anchor to two N atoms in the graphitic carbon nitride support, but does X-ray absorption give any evidence for Fe-N bond formation? I'm not sure if Fe-N and Fe-O can be differentiated. If Fe₂ is anchored on O atoms, what would the results be?*

Reply: Thank you for your helpful suggestion. Herein, it is necessary to claim that EXAFS cannot distinguish the coordinated N and O atoms, because they give similar scattering parameters due to their neighboring positions in the periodic table of elements. The coordination environment for Fe₂/mpg-C₃N₄ was investigated by quantitative least-squares EXAFS curve-fitting, as we had described in the manuscript. In the EXAFS fitting, three scattering paths (Fe-N(O)₁, Fe-N(O)₂ and Fe-Fe) were applied (**Supplementary Fig. S13**). The best EXAFS fitting result clearly showed that the main peak at R space derived from Fe-N(O) coordination, while the satellite peak at 2.27 Å is from Fe-Fe scattering. The corresponding structure parameters had been listed in the SI.

Figure 13: EXAFS fitting curves for Fe₂/mpg-C₃N₄.

In order to further identify the local structure of Fe₂/mpg-C₃N₄, XANES simulations, which are very sensitive to the 3D arrangement of atoms around the photo-absorber, were carried out at the Fe K-edge. **Supplementary Fig. S14-15** show that the simulated XANES spectrum based on our model agrees well with the experimental results. A series of other possible structures were also considered, but the comparison between the simulated spectra and the experimental XANES results is quite unsatisfactory (**Supplementary Fig. S14-15**). Combining the EXAFS fitting and the

XANES simulations, the atomic structure of the Fe₂/mpg-C₃N₄ sample can be well revealed.

Figure S14: Comparison between the Fe K-edge XANES experimental spectrum (solid red line) and the theoretical spectrum (solid blue line) calculated with the inset structure.

Figure S15: Comparison between the Fe K-edge XANES experimental spectrum (solid red line) and the theoretical spectrum (solid blue line) calculated with others different structures.

The Fe K-edge theoretical XANES calculations were carried out with the FDMNES code in the framework of real-space full multiple-scattering (FMS) scheme using Muffin-tin approximation for the potential (*Rev. Mod. Phys.* **72**, 621–654 (2000); Joly, Y. X-ray absorption near-edge structure calculations beyond the muffin-tin approximation. *Phys. Rev. B* **63**, 125120 (2001); Bunău, O. & Joly, Y. Self-consistent aspects of X-ray absorption calculations. *J. Phys. Condens. Matter.* **21**, 345501 (2009)). The energy dependent exchange-correlation potential was calculated in the real Hedin-Lundqvist scheme, and then the spectra convoluted using a Lorentzian function with an energy-dependent width to account for the broadening due both to the core-hole width and to the final state width.

The mpg-C₃N₄ support can provide abundant N atoms as anchoring sites to stabilize the Fe₂ clusters. The interaction between mpg-C₃N₄ and O atoms, however, is very weak unless defects on the substrate are explicitly involved. Thus, the possibility of anchoring Fe₂ on O is expected to be very low.

4. *I believe the m-C₃N₄ support is also important, though itself does not have high activity in the reaction. Can the authors do a comparison with graphene or GO with Fe₂ deposited as in m-C₃N₄?*

Reply: We thank the reviewer for the very good advice. We have tried to synthesize the Fe₂ clusters on the graphene oxide (GO). When this support was used, the Fe₂ clusters agglomerated to Fe nanoparticles (**Supplementary Fig. S6-7**) during the process of thermal decomposition. It is due to a lack of N atoms that can anchor the Fe₂ clusters. The same results are expected when graphene is used. The above results demonstrate the importance of the mpg-C₃N₄ support in the synthesis of the Fe₂ catalysts.

In a typical synthesis of Fe/GO, 2mg Bis(dicarbonylcyclopentadienyliron) ($\text{Fe}_2\text{O}_4\text{C}_{14}\text{H}_{10}$) and GO (200mg) were dissolved in the DMF (100 mL) under stirring at room temperature for 24h. The product was separated by centrifugation and washed subsequently with DMF then methanol thoroughly. The as-prepared powder was transferred into a ceramic both and then placed into a tube furnace maintaining 300 °C for 2h under flowing mixture of 5% H_2/Ar atmosphere with a heating rate of 5 °C min^{-1} . The Fe loading is 0.17% determined by ICP-AES analysis.

Figure S6. HAADF-STEM images of the product using the GO as support.

Figure S7. Corresponding element maps showing distributions of Fe (green), N (red), C (blue), respectively.

Reply to the Report of Reviewer 3

Referee:

Small metal clusters mounted on solid supports are the subjects of growing attention and the focus of quite a lot of catalysis research.

The subject manuscript is another example, this one with iron clusters and a catalytic epoxidation reaction.

The work could be improved in a number of respects.

Reply: We appreciate the Referee's comments that we address below point by point.

- 1. The identification of the iron clusters by STEM and XAFS should be stronger. The images show clusters of various sizes. The images should be subjected to statistical analysis of the sizes and the identification of the clusters as dimers should be examined critically.*

Reply: We thank the reviewer for the very good advice. The statistical analysis on 100 pairs of Fe₂ dimers shows that the projected Fe–Fe distance between adjacent bright dots varies from 1.20 to 2.45 Å (**Supplementary Fig. S5**). The largest value, 2.45 Å, is consistent with the bond length of a Fe₂ dimer. Since the AC HAADF-STEM image represents a two-dimensional projection along the incident beam direction, the detailed features of Fe₂ clusters are different from each other depending on their orientations in three dimensions. Bright twin dots with a distance of 2.45 Å are assigned to the Fe₂ clusters parallel to the support. Meanwhile, bright twin dots with a distance of less than 2.45 Å are assigned to the Fe₂ clusters not parallel to the support.

Figure S5. Statistical Fe–Fe distance in the observed Fe₂ clusters

2. *The XAFS data analysis should include fits with various structural models and a statistically based selection of the best fitting model and a critical assessment of the comparison.*

Reply: Thank you for the helpful suggestion. The coordination environment for Fe₂/mpg-C₃N₄ was investigated by quantitative least-squares EXAFS curve-fitting, as we had described in the manuscript. In the EXAFS fitting, three scattering paths (Fe-N(O)₁, Fe-N(O)₂ and Fe-Fe) were applied (**Supplementary Fig. S13**). The best EXAFS fitting result clearly showed that the main peak at R space derived from Fe-N(O) coordination, while the satellite peak at 2.27 Å is from Fe-Fe scattering. The corresponding structure parameters had been listed in the SI.

In order to further identify the local structure of Fe₂/mpg-C₃N₄, XANES simulations, which are very sensitive to the 3D arrangement of atoms around the photo-absorber, were carried out at the Fe K-edge. **Supplementary Fig. S14-15** show that the simulated XANES spectrum based on our model agrees well with the experimental results. A series of other possible structures were also considered, but the comparison between the simulated spectra and the experimental XANES results is quite unsatisfactory (**Supplementary Fig. S14-15**). Combining the EXAFS fitting and the

XANES simulations, the atomic structure of the Fe₂/mpg-C₃N₄ sample can be well revealed.

Figure S13. EXAFS fitting curves for Fe₂/mpg-C₃N₄.

Figure S14. Comparison between the Fe K-edge XANES experimental spectrum (solid red line) and the theoretical spectrum (solid blue line) calculated with the inset structure.

Figure S15. Comparison between the Fe K-edge XANES experimental spectrum (solid red line) and the theoretical spectrum (solid blue line) calculated with others different structures.

The Fe K-edge theoretical XANES calculations were carried out with the FDMNES code in the framework of real-space full multiple-scattering (FMS) scheme using Muffin-tin approximation for the potential (*Rev. Mod. Phys.* **72**, 621–654 (2000); Joly, Y. X-ray absorption near-edge structure calculations beyond the muffin-tin approximation. *Phys. Rev. B* **63**, 125120 (2001); Bunău, O. & Joly, Y. Self-consistent aspects of X-ray absorption calculations. *J. Phys. Condens. Matter.* **21**, 345501 (2009)). The energy dependent exchange-correlation potential was calculated in the real Hedin-Lundqvist scheme, and then the spectra convoluted using a Lorentzian function with an energy-dependent width to account for the broadening due both to the core-hole width and to the final state width.

3. *The assumption that the groups on the iron are removed in the treatment and not replaced by something else is glib. These are reactive species.*

Reply: In the previous manuscripts, we used the IR and TGA to indicate a complete removal of the ligands. In this manuscripts, we further confirm the ligands have been removed completed by the EXAFS. The Fourier-transformed (FT) k^3 -weighted EXAFS spectrum of the Fe_2 precursor is shown in Supplementary Fig. S8. Here, a peak at a high R value (ca. 2.50 Å) corresponds to the Fe-Fe coordination path. The other two peaks, at low R values (ca. 1.82 and 2.14 Å), are assigned to the ligands of the Fe_2 precursor. Interestingly, these two peaks disappeared in the spectra of the $\text{Fe}_2/\text{mpg-C}_3\text{N}_4$ samples (main text Fig. 2b and 2c), meaning that the ligands had been completely removed. The final structure of $\text{Fe}_2/\text{mpg-C}_3\text{N}_4$ is determined by EXAFS, and atomistic simulations. We optimized numerous structure based on fitting various result according to the XANES, and final concluded the structure reported in the text is the best result.

Figure S8. The Fourier transform (FT) of EXAFS spectra at the Fe k-edge of Fe_2 precursor.

4: The catalytic data would have more meaning if they were compared with catalytic data for a good known epoxidation catalyst. The stability of the catalyst should be tested with flow reactor operation for a long enough period to allow a good assessment.

Reply: We thank the reviewer for the very good advice. We have compared the performance of the Fe_2 sample with that of other known catalysts. Noble metal nanoparticles (such as Ru, Rh, Pd, Au and Pt) have been known to be good catalysts

for epoxidation (*Appl. Catal. A: Gen.* 514, 253 (2016); *ACS Catal.* 7, 3483 (2017)). Herein, we synthesized a series of noble metal nanoparticles (**Supplementary Fig. S21-25**). **Supplementary Table S2** shows that the activity of Fe₂/mpg-C₃N₄ is much better than all the metal nanoparticles.

Table S2: Catalytic epoxidation of *trans*-stilbene by Fe₂/mpg-C₃N₄ and different noble metal nanoparticles catalysts.

Catalyst	Conversion (%)	Selectivity (%)
Fe ₂ /mpg-C ₃ N ₄	91	93
Au NPs/mpg-C ₃ N ₄	35	83
Ru NPs/mpg-C ₃ N ₄	28	87
Rh NPs/mpg-C ₃ N ₄	trace	-
Pd NPs/mpg-C ₃ N ₄	trace	-
Pt NPs/mpg-C ₃ N ₄	17	90

Figure S21. TEM image of Ru nanoparticles.

Figure S22. TEM image of Rh nanoparticles.

Figure S23. TEM image of Pd nanoparticles.

Figure S24. TEM image of Pt nanoparticles.

Figure S25. TEM image of Au nanoparticles.

We have tried our best to cooperate with Institute of Process Engineering, Chinese Academy of Science to conduct the flow reaction experiment. The appliance of the flow reaction experiment (**Figure R2**) consists of oxygen canister (A), injection pump (B), preheating oven (C), reacting furnace (D), condenser pipe (E) and collector (F). When we tested the performance of $\text{Fe}_2/\text{mpg-C}_3\text{N}_4$ catalyst based on the flow reactor operation (oxygen gas flow rate: 50 mL/min; catalyst: 200 mg; liquid flow rate: 0.05

mL/min), only 41% of conversion was obtained (**Figure R3**). The activity based on the flow reactor operation is lower than the activity based on test tube, which may be attributed to the insufficiency contact between the catalysts and the reactants. To further demonstrate the stability of $\text{Fe}_2/\text{mpg-C}_3\text{N}_4$ catalyst, we also extended the times of the recycling. After fifteen cycles, the $\text{Fe}_2/\text{mpg-C}_3\text{N}_4$ sample still maintains its pore structure and exhibits robust recycling capability with well-retained activity and selectivity (**Main Text Fig. 3b**). The unchanged structures as fresh samples identified by HAADF-STEM and AC HAADF-STEM images further corroborate the stability of the catalyst (**Supplementary Fig. S26**).

Figure R2: The flow reactor operation (oxygen canister (A), injection pump (B),

preheating oven (C), reacting furnace (D), condenser pipe (E) and collector (F))

Figure R3: Recycle of $\text{Fe}_2/\text{mpg-C}_3\text{N}_4$ for catalytic epoxidation of trans-stilbene under flow reactor operation.

Figure 3b: Recycle of $\text{Fe}_2/\text{mpg-C}_3\text{N}_4$ for catalytic epoxidation of trans-stilbene.

Figure S26. a) HAADF-STEM images of Fe₂/mpg-C₃N₄ after 15 times recycle. b) corresponding element maps showing distributions of Fe (green), N (red), C (blue), respectively. c) AC HAADF-STEM images of Fe₂/mpg-C₃N₄. d) Magnified AC HAADF-STEM images of Fe₂/mpg-C₃N₄ after 15 times recycle.

Reviewers' comments:

Reviewer #1 (Remarks to the Author):

The raised questions and comments from the original manuscript have been fully addressed by the authors. I recommend to publish this beautiful work without further revision.

Reviewer #2 (Remarks to the Author):

The authors have carried out additional analyses and my questions are basically addressed except the local structure of Fe₂/C₃N₄, i.e. Fe-O or Fe-N bonding, but I agree with them that it is a very difficult task. The supports of catalysts are often highly defective, so the defect modes may not be ignored. The R1 manu is publishable.

Reviewer #3 (Remarks to the Author):

This reviewer's earlier reservations remain. The TEM images are not sufficient to exclude mixtures and do not support the authors' conclusions about dimeric iron species on the substrate. The XAFS analysis still does not provide a full, statistically sound comparison of various structure models.

The structural conclusions are overstated.

In this reviewer's judgement the paper falls short of the rigor expected for publication in the journal.

Response to the Referees' Comments

Reply to the Report of Reviewer 1

Referee:

The raised questions and comments from the original manuscript have been fully addressed by the authors. I recommend to publish this beautiful work without further revision.

Reply: We thank the reviewer for the positive comments on the revised manuscript.

Reply to the Report of Reviewer 2

Referee:

The authors have carried out additional analyses and my questions are basically addressed except the local structure of Fe_2/C_3N_4 , i.e. Fe-O or Fe-N bonding, but I agree with them that it is a very difficult task. The supports of catalysts are often highly defective, so the defect modes may not be ignored. The R1 manu is publishable.

Reply: We thank the reviewer for the positive comments.

Reply to the Report of Reviewer 3

Referee:

This reviewer's earlier reservations remain. The TEM images are not sufficient to exclude mixtures and do not support the authors' conclusions about dimeric iron species on the substrate. The XAFS analysis still does not provide a full, statistically sound comparison of various structure models.

The structural conclusions are overstated. In this reviewer's judgement the paper falls short of the rigor expected for publication in the journal.

Reply: We thank the reviewer for the comments that we address below point by point.

To further illustrate the Fe₂ site, we reduced the loading amount of Fe₂. AC-STEM images shows that the Fe atoms in the spherical electron microscope were still present as Fe₂ clusters, indicating that the Fe₂ clusters did not decompose into single atoms during the synthetic process. Since the AC HAADF-STEM image represents a two-dimensional projection along the incident beam direction, the detailed features of Fe₂ clusters are different from each other depending on their orientations in three dimensions. We also performed TOF-SIMS characterization of the samples. The data shows that there is only Fe₂, but no larger Fe clusters, such as Fe₃ or Fe₄, indicating that the Fe₂ clusters did not agglomerate. Additionally, a series of other possible structures (such as Fe₁N_mC_n, Fe₂N_mC_n, Fe₂ON_mC_n and Fe₂O₂N_mC_n) were also considered, but the comparison between the simulated spectra and the experimental EXAFS and XANES results is unsatisfactory, further confirming the reported structure is the most likely actual structure.

Figure S8. AC HAADF-STEM images of $\text{Fe}_2/\text{mpg-C}_3\text{N}_4$. (left: the loading Fe is 0.025wt%; right: the loading Fe is 0.012wt%).

Figure S9. The TOF-SIMS spectrum around m/z 111.9 signals of $\text{Fe}_2/\text{mpg-C}_3\text{N}_4$, Fe_2 precursor and $\text{mpg-C}_3\text{N}_4$. The obvious fragment signal in $\text{Fe}_2/\text{mpg-C}_3\text{N}_4$ was detected at m/z 111.88 in positive ion mode, which was consistent with observation of Fe_2^+ from Fe_2 precursor. In addition, the m/z 111.88 signal (Fe_2^+) of $\text{mpg-C}_3\text{N}_4$ sample was not detected. The results were a powerful proof to further confirm the presence of Fe_2 cluster species in $\text{Fe}_2/\text{mpg-C}_3\text{N}_4$ sample.

Figure S10. The TOF-SIMS spectrum around Fe_3^+ and Fe_4^+ signals of $\text{Fe}_2/\text{mpg-C}_3\text{N}_4$, and $\text{mpg-C}_3\text{N}_4$. No fragment signals in $\text{Fe}_2/\text{mpg-C}_3\text{N}_4$ was detected at around m/z of Fe_3^+ (167.80) and Fe_4^+ (223.74). The results were a powerful proof to further confirm no larger Fe clusters species, such as Fe_3 or Fe_4 , in $\text{Fe}_2/\text{mpg-C}_3\text{N}_4$ sample.

Figure S17. Comparison between the Fe K-edge EXAFS experimental spectrum (solid black line) and the theoretical spectrum (solid red line) calculated with others different structures.

Figure S19. Comparison between the Fe K-edge XANES experimental spectrum (solid red line) and the theoretical spectrum (solid blue line) calculated with others different structures.